# Thyroid Cancer in Patients with Metabolic Syndrome or Its Components: A Nationwide Population-Based Cohort Study

**DOI:** 10.3390/cancers14174106

**Published:** 2022-08-25

**Authors:** Jae Hyun Park, Hyun Seok Cho, Jong Ho Yoon

**Affiliations:** 1Department of Surgery, Yonsei University Wonju College of Medicine, Wonju 26493, Korea; 2Department of Radiology, Yonsei University Wonju College of Medicine, Wonju 26493, Korea

**Keywords:** thyroid carcinoma, metabolic syndrome, risk factor, cohort study

## Abstract

**Simple Summary:**

The rapidly increasing coincidence of thyroid cancer and metabolic syndrome (MS) in recent decades suggests an association between the two disorders. To investigate this association, we conducted a nationwide study of a large-scale patient cohort. Between 2009 and 2011, data were collected by the Korean National Health Insurance Service for 4,658,473 persons aged 40–70 years without thyroid cancer. The incidence of thyroid cancer per 10,000 person-years in individuals with MS was significantly higher in men (6.2, *p* < 0.001) and women (21.3, *p* < 0.001) compared to those without MS. Additionally, the risk of thyroid cancer increased significantly with an increasing number of MS components even in individuals with only one or two MS components. MS and its components were significantly associated with increased risk of developing thyroid cancer.

**Abstract:**

The rapidly increasing coincidence of thyroid cancer and metabolic syndrome (MS) in recent decades suggests an association between the two disorders. To investigate this association, we conducted a nationwide study of a large-scale patient cohort. Between 2009 and 2011, data were collected by the Korean National Health Insurance Service for 4,658,473 persons aged 40–70 years without thyroid cancer. During the six-year follow-up period, participants were monitored for the development of thyroid cancer. The relative risks and incidences of thyroid cancer were calculated using multivariate Cox proportional hazards regression analyses after adjusting for age and body mass index. The risk of thyroid cancer was significantly elevated in men and women with MS or MS components, except for hyperglycaemia (*p* = 0.723) or hypertriglyceridemia (*p* = 0.211) in men. The incidence of thyroid cancer per 10,000 person-years in individuals with MS was significantly higher in men (6.2, *p* < 0.001) and women (21.3, *p* < 0.001) compared to those without MS. Additionally, the risk of thyroid cancer increased significantly with an increasing number of MS components even in individuals with only one or two MS components. MS and its components were significantly associated with increased risk of developing thyroid cancer.

## 1. Introduction

The global incidence of thyroid cancer has increased steeply over the past several decades [1,2]. Although improved detection of small subclinical thyroid cancer has contributed to this rise in incidence [3], the effect of other modifiable risk factors cannot be ignored [4]. The development of thyroid cancer is multifaceted and may be associated with other diseases or syndromes; however, the exact pathophysiologic mechanisms are not well established.

Over the same period, the prevalence of obesity and unhealthy metabolic conditions, such as metabolic syndrome (MS), has markedly increased and may be associated with the rising incidence of thyroid cancer. Indeed, MS and insulin resistance (IR) are known to increase the incidence of other cancers, including endometrial, lung, pancreatic, hepatocellular, prostate, and colorectal cancers [5,6,7,8,9,10]. Additionally, IR, obesity, and type 2 diabetes are more prevalent in patients with differentiated thyroid carcinoma than in those without [11]. Previous studies investigating the association between various metabolic conditions and the development of thyroid cancer focused on body mass index (BMI) and IR [12], but the results were limited and did not establish a clear connection between metabolic conditions and thyroid cancer.

Here, we report a large-scale, nationwide, population-based, cohort study investigating the association between MS or its components and the risk of thyroid cancer.

## 2. Materials and Methods

### 2.1. Study Population

The present study utilized the South Korean population database provided by the National Health Insurance Service (NHIS), which is the public medical insurance system that covers up to 97% of the South Korean population and provides all insured persons with standardized health screening tests every two years. The data from the NHIS encompass demographic, anthropometric, and laboratory data, medical treatments, procedures, including surgeries and diagnostic codes based on the International Classification of Disease-10th revision-Clinical Modification (ICD-10-CM).

The complete NHIS cohort has been previously described [13]. From 2009 to 2011, 8,922,940 people aged 20 or older underwent a health examination provided by NHIS. Participants meeting the following criteria were excluded from this study: history of cardiovascular events before 2011 (*n* = 441,337), participation in less than two health examinations between 2009 and 2011 (*n* = 1,134,965), aged < 40 years or > 70 years (*n* = 2,636,776), prior diagnosis of thyroid cancer before 2011 (*n* = 41,764), and those with any missing or inappropriate data (*n* = 9625). In total, 4,658,473 people (2,468,996 men and 2,189,477 women) were enrolled in the study and followed until the date of diagnosis with thyroid cancer or last follow-up by 31 December 2017 (Figure 1).

This study was conducted based on the ethical principles outlined in the Declaration of Helsinki. The study protocol was approved by the Institutional Review Board of Yonsei University Wonju College of Medicine (IRB number: CR 318349). The requirement for written informed consent was waived because anonymous and de-identified information was used for the analysis, and there was no intervention.

### 2.2. Health Examination

The health examinations provided by NHIS measured participant height, weight, BMI (calculated by dividing weight [kg] by height [m] squared), waist circumference, systolic and diastolic blood pressure (measured in a seated position after resting for a minimum of five minutes), and laboratory tests such as blood tests and urinalysis. Blood tests were performed after overnight fasting to measure serum levels of glucose, total cholesterol, triglyceride, high-density lipoprotein (HDL)-cholesterol, low-density lipoprotein (LDL)-cholesterol, and creatinine. Co-morbidities, including hypertension, diabetes mellitus, and dyslipidaemia, were indicated by ICD-10-CM codes or by claims for the agents prescribed for co-morbidities.

### 2.3. Definitions of MS and the Development of Thyroid Cancer

MS was defined according to guidelines of the American Heart Association and the National Heart, Lung, and Blood Institute, and the Asian-specific waist circumference was adopted to define abdominal obesity [14]. Briefly, individuals with three or more of the following components were diagnosed with MS: abdominal obesity (waist circumference ≥ 90 cm for males and ≥85 cm for females); elevated blood pressure (systolic ≥ 130 mmHg and/or diastolic ≥ 85 mmHg or treatment with antihypertensive medication); hyperglycaemia (fasting plasma glucose ≥ 100 mg/dL or treatment with antidiabetic agents or insulin); low HDL-cholesterol levels (<40 mg/dL for males and <50 mg/dL for females), and hypertriglyceridemia (triglyceride ≥ 150 mg/dL or treatment with lipid-lowering medication). The development of thyroid cancer was identified in hospitalization records with the ICD-10-CM code C73. Individuals who were hospitalized for thyroid cancer between 2009 and 2011 were excluded from the study.

### 2.4. Statistical Analyses

Continuous variables are presented as the mean ± standard deviation, and categorical variables are presented as percentages and absolute numbers. Baseline characteristics were compared using the chi-square test for categorical variables and the independent *t*-test for continuous variables. The incidence of thyroid cancer was calculated by dividing the number of thyroid cancer cases by 10,000 person-years. Multivariate Cox proportional hazard regression analyses were used to investigate the association of MS and its components with the risk of thyroid cancer by sex and to assess the impact of the number of MS components on the risk of thyroid cancer. Relative risks (RRs) with 95% confidence intervals (CIs) were assessed. These models were adjusted for age and BMI. Two-sided *p* values less than 0.05 were considered statistically significant. All statistical analyses were performed using STATA version 14 (StataCorp LP, College Station, TX, USA).

## 3. Results

### 3.1. Baseline Characteristics of the Study Population

The baseline characteristics of the study participants are summarized in Table 1. Within the study population, 1,012,782 (21.7%) participants had MS. During the 6-year follow-up period, 47,325 (1.0%) participants were newly diagnosed with thyroid cancer. Those who developed thyroid cancer were significantly younger (51.1 ± 7.5 years vs. 51.5 ± 8.1 years, *p* < 0.001) and significantly more were female (76.4% vs. 23.6%, *p* < 0.001) compared to those without thyroid cancer. Additionally, the mean BMI and the proportion of individuals with BMI greater than 25 was significantly higher in the thyroid cancer group than in the non-thyroid cancer group (Table 1).

Although the prevalence of MS did not differ between those with thyroid cancer and those without (*p* = 0.616), the majority of MS components did vary between the groups. Specifically, abdominal obesity and low HDL-cholesterol were significantly higher in thyroid cancer patients, whereas hyperglycaemia and hypertriglyceridemia were significantly lower in thyroid cancer patients (Table 1). In fact, the only MS component that did not vary significantly between the two groups was elevated blood pressure (*p* = 0.324) (Table 1).

### 3.2. Association of MS and Its Components with the Risk of Thyroid Cancer by Sex

The incidence of thyroid cancer per 10,000 person-years and the RRs of thyroid cancer were calculated after adjusting for age and BMI. MS was significantly associated with an increased risk of thyroid cancer in both men (*p* < 0.001, Table 2) and women (*p* < 0.001, Table 3). The incidence of thyroid cancer per 10,000 person-years was 6.2 in men with MS and 21.3 in women with MS, which was significantly higher than the incidence in men (5.2) and women (19.6) without MS (Table 2 and Table 3). Interestingly, the association of each MS component and the risk of thyroid cancer differed between males and females. All MS components were significantly associated with the risk of developing thyroid cancer in women, whereas hyperglycaemia (*p* = 0.723) and hypertriglyceridemia (*p* = 0.211) were not significantly associated with thyroid cancer risk in men (Table 2 and Table 3). In both sexes, abdominal obesity, elevated blood pressure, and low HDL-cholesterol had greater RRs for developing thyroid cancer than the other MS components. In men, abdominal obesity showed the highest RR for developing thyroid cancer (RR, 1.34 [CI, 1.29–1.40]; *p* < 0.001), followed by low HDL-cholesterol, and then elevated blood pressure. In contrast, women had the highest RR for developing thyroid cancer with low HDL-cholesterol (RR, 1.19 [CI, 1.16–1.22]; *p* < 0.001), followed by elevated blood pressure, and then abdominal obesity. In general, the incidence of thyroid cancer per 10,000 person-years in women with MS or with individual MS components was higher than in men (Table 2 and Table 3).

### 3.3. Association of the Number of Metabolic Syndrome Components with the Risk of Thyroid Cancer

Although MS was defined by the presence of three or more MS components, we wanted to assess the contribution of each additional MS component to the risk of thyroid cancer. We found that higher numbers of MS components resulted in significantly higher RRs for developing thyroid cancer in men and women (Table 4). This was true even in participants with only one or two MS components. Individuals with all five MS components showed 53% (RR, 1.53 [CI, 1.31–1.80]; *p* < 0.001) and 43% (RR, 1.43 [CI, 1.31–1.56]; *p* < 0.001) higher RRs for thyroid cancer in men and women, respectively, than those without any MS components (Table 4). The incidence of thyroid cancer per 10,000 person-years, which was adjusted for age and BMI, also gradually increased with the number of MS components and was much higher in women than in men (Table 4).

## 4. Discussion

This large-scale, nationwide, population-based, cohort study of 4,658,473 persons aged 40–70 years demonstrated that the risk of developing thyroid cancer was significantly associated with the presence of MS and its components. This study also revealed a considerable role for sex in the risk and prevalence of thyroid cancer. In women, all five MS components were significantly linked to the risk of thyroid cancer, whereas only abdominal obesity, low HDL-cholesterol, and elevated blood pressure increased the risk of thyroid cancer in men. Moreover, the incidence of thyroid cancer per 10,000 person-years in subjects with MS or significant MS components was substantially higher in women than in men. Another key outcome of this study was that the risk of thyroid cancer increased significantly with the number of MS components. Indeed, the RR and incidence of thyroid cancer were significantly higher in subjects with just one or two MS components who did not meet the full definition of MS than in those without any MS components. Altogether, this study establishes an important association between MS and its components with the risk of thyroid cancer, which has important clinical implications for thyroid cancer surveillance in individuals with even one MS component.

Most previous studies have investigated the association between BMI or IR and the development of thyroid cancer. One systemic review of 42 articles revealed that patients with IR (RR, 1.59 [CI, 1.12–2.27]; *p* = 0.01), dysglycaemia (RR, 1.40 [CI, 1.15–1.70]; *p* < 0.001), BMI > 25 kg/m^2^ (RR, 1.35 [CI, 1.23–1.48]; *p* < 0.001), and hypertension (RR, 1.34 [CI, 1.22–1.47]; *p* < 0.001) showed an increased risk for thyroid cancer, whereas those with dyslipidaemia did not [12]. In the same review, the authors assigned quality levels to the evidence provided for each factor, ranging from ‘very low’ to ‘very high’. Importantly, the evidence quality levels were ‘low’ or ‘very low’ for every metabolic parameter except IR and BMI, which were graded ‘moderate’ [12]. Thus, these results are not sufficient to establish an association between metabolic status and the risk of thyroid cancer. Similarly, there was limited evidence regarding the impact of MS on the risk of developing thyroid cancer. To our knowledge, only one other study (15) has comprehensively evaluated the association between MS or its components and the risk of developing thyroid cancer in a large-scale patient cohort. In that study, 9,890,917 South Korean adults without thyroid cancer were grouped according to obesity status and assessed [15]. The study demonstrated that thyroid cancer risk was higher in the MS group (Hazard ratio [HR], 1.15 [CI, 1.13–1.17]) than in the non-MS group, and the association between MS and thyroid cancer risk was significant in the obese group (defined as BMI > 25 kg/m^2^) (HR, 1.10 [CI, 1.07–1.13]) but not in the non-obese group (HR, 1.00 [CI, 0.98–1.03]) [15]. The study also emphasized the combined effect of obesity and MS on the risk of developing thyroid cancer, revealing that obese men with MS had the highest risk of thyroid cancer, but obese women with MS did not [15]. Additionally, they showed that the risk of thyroid cancer significantly increased with an increase in the number of MS components in the obese group, but not in the non-obese group [15]. Although the study supports an association between MS and thyroid cancer, it suffers from an analytical perspective. The authors analysed the HRs of MS and its components for thyroid cancer development according to obesity status to obtain their results. However, BMI, which was used to define obesity status, is a continuous variable. To evaluate the association between a continuous variable and a target outcome, continuous data, like BMI, are dichotomized, split into several categories, or handled as a continuous variable with an imposed linear relationship between the variable and the outcome [16]. These commonly used modelling approaches carry relatively stringent statistical assumptions and can lead to loss of information, which may either weaken the predictive ability of the model or yield a poor fit between the continuous variable and the target outcome [16]. Thus, these approaches are not recommended in fields of medicine with variables and outcomes that show a non-linear continuous association [17]. The present study addresses this concern by adjusting for age and BMI prior to evaluation of risk.

Unlike earlier studies, the present study enrolled only subjects aged 40–70 years. This range was selected because thyroid cancer and MS are more prevalent at this age, and atypical events tend to occur in extremely young or old individuals. Although the study discussed above compared baseline characteristics between MS and non-MS groups, our study compared thyroid cancer and non-thyroid cancer groups, because thyroid cancer development was the target outcome in our study. All RRs and incidences of thyroid cancer were calculated after adjusting for age and BMI to avoid any statistical inappropriateness and to enhance the applicability of our results. Interestingly, the RRs and incidences of thyroid cancer gradually increased with the number of MS components in both men and women as well as in individuals with only one or two MS components who did not meet the definition for MS. Altogether, these results further support the association between MS and the risk of thyroid cancer.

Several pathophysiological mechanisms have been proposed to explain how thyroid carcinogenesis is associated with MS or its components. For example, thyroid carcinogenesis may be driven by increased proliferation, angiogenesis, cellular mobility, and DNA damage due to IR and hyperglycaemia. IR may also overstimulate insulin-like growth factor-1, -2, and insulin receptors, which contribute to thyroid carcinogenesis. Additionally, obesity is associated with the release of cytokines, bioactive molecules, and oestrogen from visceral fat, immune cells, and adipocytes, which are also factors in thyroid cancer development [18,19,20,21,22,23]. These proposed mechanisms all revolve around IR and obesity, which do not fully encompass MS. Although IR and obesity are key factors in the pathophysiology of MS, the impact of other MS components on thyroid carcinogenesis remains unclear. Based on our results, abdominal obesity, low HDL-cholesterol, and elevated blood pressure impose greater RRs for developing thyroid cancer than hyperglycaemia and hypertriglyceridemia, which were not significant in men. Nevertheless, the exact pathophysiological mechanisms remain unclear. As correlation does not equal causation, it is important to identify whether the association of MS or its components and the development of thyroid cancer is due to the incidental overlapping of two prevalent disorders or due to connected pathophysiological mechanisms. To make this distinction, additional large-scale, multicentre-based studies are needed. If these two disorders are indeed mechanistically connected, then MS and its components would not only serve as important risk factors for thyroid cancer development but would also provide a clinically actionable prevention strategy via modification of lifestyle behaviours, such as physical exercise and diet.

In this study, we selected patients aged 40–70 years and excluded data that were not mathematizable or subjective, such as smoking status, alcohol consumption status, and physical activity, from the analyses. We focused instead on concise analyses of basic demographics (age and sex), anthropometric and laboratory measurements, and the resultant statuses of MS and its components. Of note, these decisions may have introduced a selection bias in the study population due to the non-obligatory nature of the health examinations provided by the NHIS. We also acknowledge a lack of clinicopathological data specific to thyroid cancer, including histologic subtype, primary tumor size, extrathyroidal extension, lymph node or distant metastasis, which reflect the biological aggressiveness of cancer, and thyroid functional status, its autoimmunity, and inflammatory status. Furthermore, our study could not include the duration of MS or its components in the analyses.

## 5. Conclusions

Based on the results of the present study, MS and its components, especially abdominal obesity, low HDL-cholesterol, and elevated blood pressure, are likely to contribute to the development of thyroid cancer, with a higher incidence in women than in men. Although the exact mechanisms that connect the two disorders remain unclear, MS and its components are predictive of thyroid cancer risk and may be useful in screening efforts to diagnose thyroid cancer at early stages.

## Figures and Tables

**Figure 1 cancers-14-04106-f001:**
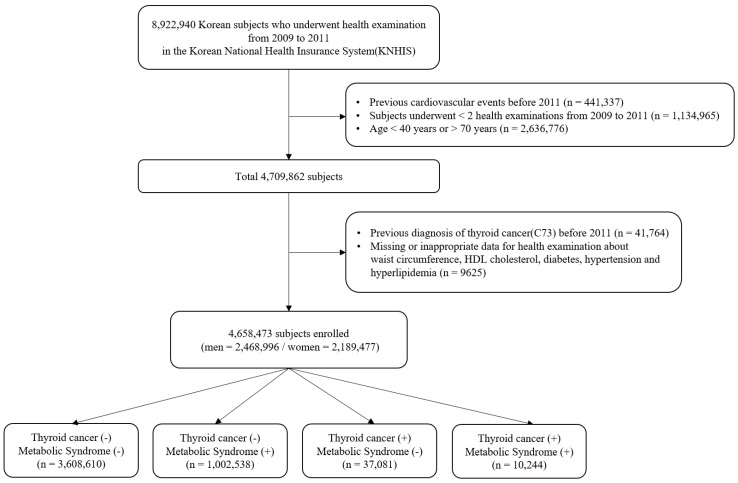
Flowchart of participants’ inclusion and exclusion.

**Table 1 cancers-14-04106-t001:** Baseline characteristics of the study population.

Characteristics	Total * (*n* = 4,658,473)	Thyroid Cancer Group * (*n* = 47,325)	Non-Thyroid Cancer Group * (*n* = 4,611,148)	*p* Value
Age, years (mean ± SD)	51.5 ± 8.1	51.1 ± 7.5	51.5 ± 8.1	<0.001
Sex				<0.001
Male	2,468,996 (53.0)	11,168 (23.6)	2,457,828 (53.3)	
Female	2,189,477 (47.0)	36,157 (76.4)	2,153,320 (46.7)	
BMI (mean ± SD)	24.0 ± 3.0	24.2 ± 3.0	24.0 ± 3.0	<0.001
<18	44,468 (1.0)	279 (0.6)	44,189 (1.0)	0.063
18 to <23	1,729,714 (37.1)	17,143 (36.2)	1,712,571 (37.1)	<0.001
23 to <25	1,299,606 (27.9)	13,137 (27.8)	1,286,469 (27.9)	0.009
25 to <30	1,452,751 (31.2)	14,943 (31.6)	1,437,808 (31.2)	<0.001
≥30	131,221 (2.8)	1811 (3.8)	129,410 (2.8)	<0.001
Metabolic syndrome				0.616
No	3,645,691 (78.3)	37,081 (78.4)	3,608,610 (78.3)	
Yes	1,012,782 (21.7)	10,244 (21.6)	1,002,538 (21.7)	
Metabolic syndrome components				
Abdominal obesity				<0.001
No	3,824,244 (82.1)	38,235 (80.8)	3,786,009 (82.1)	
Yes	834,229 (17.9)	9090 (19.2)	825,139 (17.9)	
Elevated blood pressure				0.324
No	2,379,655 (51.1)	24,068 (50.9)	2,355,587 (51.1)	
Yes	2,278,818 (48.9)	23,257 (49.1)	2,255,561 (48.9)	
Hyperglycaemia				<0.001
No	3,088,048 (66.3)	33,476 (70.7)	3,054,572 (66.2)	
Yes	1,570,425 (33.7)	13,849 (29.3)	1,556,576 (33.8)	
Hypertriglyceridemia				<0.001
No	3,177,115 (68.2)	34,921 (73.8)	3,142,194 (68.1)	
Yes	1,481,358 (31.8)	12,404 (26.2)	1,468,954 (31.9)	
Low HDL-cholesterol				<0.001
No	3,785,772 (81.3)	35,098 (74.2)	3,750,674 (81.3)	
Yes	872,701 (18.7)	12,227 (25.8)	860,474 (18.7)	

SD, standard deviation; BMI, body mass index; HDL, high-density lipoprotein. * Expressed as *n* (%) unless otherwise noted.

**Table 2 cancers-14-04106-t002:** Association of metabolic syndrome or its components with the risk of thyroid cancer in male participants.

Characteristics	Number of Subjects	Number of Subjects with Thyroid Cancer	Incidence Per 10,000 Person-Years	RR (95% CI) *	*p* Value
Metabolic syndrome					<0.001
No	1,855,184	7986	5.2	Reference	
Yes	613,812	3182	6.2	1.19 (1.14–1.24)	
Metabolic syndrome components					
Abdominal obesity					<0.001
No	1,987,144	8380	5.1	Reference	
Yes	481,852	2788	6.9	1.34 (1.29–1.40)	
Elevated blood pressure					<0.001
No	1,177,665	4922	4.6	Reference	
Yes	1,291,331	6246	6.4	1.17 (1.13–1.22)	
Hyperglycaemia					0.723
No	1,465,193	6644	5.4	Reference	
Yes	1,003,803	4524	5.4	0.99 (0.96–1.03)	
Hypertriglyceridemia					0.211
No	1,443,475	6422	5.3	Reference	
Yes	1,025,521	4746	5.5	1.02 (0.99–1.06)	
Low HDL-cholesterol					<0.001
No	2,170,410	9499	5.2	Reference	
Yes	298,586	1669	6.7	1.27 (1.21–1.34)	

RR, relative risk; CI, confidence interval; HDL, high-density lipoprotein. * Adjusted for age and body mass index.

**Table 3 cancers-14-04106-t003:** Association of metabolic syndrome or its components with the risk of thyroid cancer in female participants.

Characteristics	Number of Subjects	Number of Subjects with Thyroid Cancer	Incidence Per 10,000 Person-Years	RR (95% CI) *	*p* Value
Metabolic syndrome					<0.001
No	1,790,507	29,095	19.6	Reference	
Yes	398,970	7062	21.3	1.16 (1.13–1.19)	
Metabolic syndrome components					
Abdominal obesity					<0.001
No	1,837,100	2,9855	19.6	Reference	
Yes	352,377	6302	21.5	1.15 (1.12–1.18)	
Elevated blood pressure					<0.001
No	1,201,990	19,146	19.2	Reference	
Yes	987,487	17,011	20.7	1.17 (1.15–1.20)	
Hyperglycaemia					0.008
No	1,622,855	26,832	19.8	Reference	
Yes	566,622	9325	19.9	1.03 (1.01–1.06)	
Hypertriglyceridemia					<0.001
No	1,733,640	28,499	19.8	Reference	
Yes	455,837	7658	20.2	1.06 (1.04–1.091)	
Low HDL-cholesterol					<0.001
No	1,615,362	25,599	19.1	Reference	
Yes	574,115	10,558	22.1	1.19 (1.16–1.22)	

RR, relative risk; CI, confidence interval; HDL, high-density lipoprotein. * Adjusted for age and body mass index.

**Table 4 cancers-14-04106-t004:** Association of the number of metabolic syndrome components with the risk of thyroid cancer.

Number of Metabolic Components	Number of Subjects	Number of Subjects with Thyroid Cancer	Incidence Per 10,000 Person-Years	RR (95% CI) *	*p* Value
Male					
0	479,721	1906	4.8	Reference	
1	710,554	3007	5.1	1.07 (1.01–1.13)	0.023
2	664,909	3073	5.5	1.17 (1.10–1.23)	<0.001
3	421,177	2073	5.9	1.23 (1.16–1.31)	<0.001
4	165,985	944	6.8	1.41 (1.30–1.53)	<0.001
5	26,650	165	7.4	1.53 (1.31–1.80)	<0.001
Female					
0	671,328	10,311	18.5	Reference	
1	668,745	11,017	19.8	1.13 (1.10–1.16)	<0.001
2	450,434	7767	20.7	1.23 (1.19–1.27)	<0.001
3	257,578	4490	20.9	1.27 (1.23–1.32)	<0.001
4	112,869	2027	21.5	1.33 (1.27–1.40)	<0.001
5	28,523	545	22.9	1.43 (1.31–1.56)	<0.001

RR, relative risk; CI, confidence interval. * Adjusted for age and body mass index.

## Data Availability

Some or all datasets generated during and/or analyzed during the current study are not publicly available but are available from the corresponding author on reasonable request.

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
