# Peer review of "Thyroid Cancer in Patients with Metabolic Syndrome or Its Components: A Nationwide Population-Based Cohort Study"

_cancers, 2022, doi:10.3390/cancers14174106_

Round 1

Reviewer 1 Report

The authors have undertaken a large scale population study utilising a nationwide  health assessment and clinical database in which metabolic syndrome and thyroid cancer could be assessed in the same population sample. Between 2009-2011 over 4.5 million persons were enrolled following assessment for eligibility and the relative risk and incidence of thyroid cancer and MS were determined in the population studied. The authors conclude that MS and its components are significantly associated with an increased risk of thyroid cancer.

Comments

The manuscript is well written and the strength of the paper is the very large population data that the authors have obtained from the Korean National Health Insurance Service. The authors have presented statistically significant data demonstrating  associations between MS and its components with thyroid cancer however it seems to most observers that given the high prevalence of MS amongst all populations in the world that naturally there will be an association between MS and most common cancers .

Author Response

The manuscript is well written and the strength of the paper is the very large population data that the authors have obtained from the Korean National Health Insurance Service. The authors have presented statistically significant data demonstrating associations between MS and its components with thyroid cancer however it seems to most observers that given the high prevalence of MS amongst all populations in the world that naturally there will be an association between MS and most common cancers.

R: Thank you for the insightful comments. We strongly agree with you on that point. Due to the high prevalence of MS, associations with MS and some cancers, including thyroid cancer, can be observed and have already been reported. The key outcome of our study was that the risk of thyroid cancer increased significantly with the number of MS components. Indeed, the relative risk and incidence of thyroid cancer were significantly higher even in subjects with just one or two MS components who did not meet the full definition of MS than in those without any MS components. We already described what you commented as a limitation of our study in the Discussion section (line 250-258).

Reviewer 2 Report

The Authors conducted a cohort study aiming at the investigation of the association between metabolic syndrome and the risk of thyroid cancer.

They showed that the risk of thyroid cancer increased significantly with an increasing number of metabolic syndrome components even in individuals with only one or two metabolic syndrome components. Therefore, they assessed that the metabolic syndrome and its components were significantly associated with increased risk of developing thyroid cancer.

The paper is interesting. However, some points need to be addressed:

1)    It is well known that hypothyroidism is associated with thyroid cancer. 

Have the authors evaluated the effect of TSH on thyroid cancer? If not they should at least discuss this point.

2)    It is well known that thyroid autoimmunity is a risk factor for thyroid cancer. Have the authors evaluated the association between thyroid autoimmunity and thyroid cancer? If not they should at least discuss this point.

3)    I suggest to discuss the role of an inflammatory status in the development of cancer. 

See:

-Semin Cancer Biol. 2020 Aug;64:135-146. 

-Int J Mol Sci. 2019 Sep 7;20(18):4413. 

Author Response

Point 1: The Authors conducted a cohort study aiming at the investigation of the association between metabolic syndrome and the risk of thyroid cancer. They showed that the risk of thyroid cancer increased significantly with an increasing number of metabolic syndrome components even in individuals with only one or two metabolic syndrome components. Therefore, they assessed that the metabolic syndrome and its components were significantly associated with increased risk of developing thyroid cancer. The paper is interesting. However, some points need to be addressed:

1)    It is well known that hypothyroidism is associated with thyroid cancer. Have the authors evaluated the effect of TSH on thyroid cancer? If not they should at least discuss this point.

2)    It is well known that thyroid autoimmunity is a risk factor for thyroid cancer. Have the authors evaluated the association between thyroid autoimmunity and thyroid cancer? If not they should at least discuss this point.

3)    I suggest to discuss the role of an inflammatory status in the development of cancer. 

See:

-Semin Cancer Biol. 2020 Aug;64:135-146.

-Int J Mol Sci. 2019 Sep 7;20(18):4413. 

R: Thank you for the insightful comments. The present study utilized the South Korean population database provided by the National Health Insurance Service (NHIS), which is the public medical insurance system that covers up to 97% of the South Korean population and provides all insured persons with standardized health screening tests every two years. The data from the NHIS encompasses demographics, anthropometric and laboratory data, medical treatments, procedures, including surgeries, and diagnostic codes based on the International Classification of Disease-10th revision-Clinical Modification (ICD-10-CM). Unfortunately, the laboratory tests provided by this health screening do not include a thyroid function test and tests of thyroid autoimmunity and inflammatory status like delta neutrophil index. Because of a lack of these data, we could not assess the association between the thyroid functional status, its autoimmunity, or inflammatory status and the development of thyroid cancer that the reviewer commented. We ask for your excuse. We added these as a limitation of our study in the Discussion section (line 264-268).

Reviewer 3 Report

Manuscript presents the analysis of data collected by the Korean National Health Insurance. The potential strenght of the research is a large study group. However, analyzed data are scarce. Patients with metabolic syndrome undergo investigations more often that patients without. Therefore, frequency of ultrasound examinations in this group might be the strongest confounding factor influecning achieved result. Also, the conclusion, that patients with MS should have screening tests for thyroid cancer is not based on the achieved results. Furtheremore, there are no data regarding the histopathological type of the cancer, no data about tumor staging. Therefore, in my opinion, this study does not give any clinically relevant information.

Author Response

Manuscript presents the analysis of data collected by the Korean National Health Insurance. The potential strenght of the research is a large study group. However, analyzed data are scarce. Patients with metabolic syndrome undergo investigations more often that patients without. Therefore, frequency of ultrasound examinations in this group might be the strongest confounding factor influecning achieved result.

R: Thank you for the insightful comments. We totally agree with you on that point. However, unfortunately, we could not evaluate the impact of the point that you mentioned on the increased detection rate of thyroid cancer because of limited data from these health screening tests provided by the National Health Insurance Service. We ask for you excuse.  

Also, the conclusion, that patients with MS should have screening tests for thyroid cancer is not based on the achieved results.

R: Thank you for your comment. As you mentioned, the description must have been a logical leap. We deleted it.

Furtheremore, there are no data regarding the histopathological type of the cancer, no data about tumor staging.

R: Thank you for the insightful comments. Unfortunately, these health screening tests provided by the National Health Insurance Service do not include detailed clinicopathological data of thyroid cancer. We already described what you commented as a limitation of our study in the Discussion section (line 264-268).

Round 2

Reviewer 2 Report

The Manuscript can be published in the current form.

Reviewer 3 Report

My main concern that there are no clinicopathological data and the study does not provide any relevant results can not be improved, since those data have been not collected.